# Multistep antimicrobial stewardship intervention on antibiotic prescriptions and treatment duration in children with pneumonia

Sara Rossin[1☯]*, Elisa Barbieri[2☯], Anna Cantarutti[3,4], Francesco Martinolli[1], Carlo Giaquinto[2], Liviana Da Dalt[1], Daniele Doná[2]

1 Pediatric Emergency Department, Department for Woman and Child Health, University of Padua, Padua, Italy, 2 Division of Pediatric Infectious Diseases, Department for Woman and Child Health, University of Padua, Padua, Italy, 3 Department of Statistics and Quantitative Methods, National Centre for Healthcare Research and Pharmacoepidemiology, University of Milano-Bicocca, Milan, Italy, 4 Department of Statistics and Quantitative Methods, Unit of Biostatistics, Epidemiology and Public Health, University of Milano-Bicocca, Milan, Italy

☯ These authors contributed equally to this work.
* sara.rossin1988@gmail.com

**Data Availability Statement:** All relevant data are within the manuscript and its Supporting information files.

## Abstract

### Introduction

The Italian antimicrobial prescription rate is one of the highest in Europe, and antibiotic resistance has become a serious problem with high costs and severe consequences, including prolonged illnesses, the increased period of hospitalization and mortality. Inadequate antibiotic prescriptions have been frequently reported, especially for lower respiratory tract infections (LRTI); many patients receive antibiotics for viral pneumonia or bronchiolitis or broad-spectrum antibiotics for not complicated community-acquired pneumonia. For this reason, healthcare organizations need to implement strategies to raise physicians' awareness about this kind of drug and their overall effect on the population.

The implementation of antibiotic stewardship programs and the use of Clinical Pathways (CPs) are excellent solutions because they have proven to be effective tools at diagnostic and therapeutic levels.

### Aims

This study evaluates the impact of CPs implementation in a Pediatric Emergency Department (PED), analyzing antibiotic prescriptions before and after the publication in 2015 and 2019. The CP developed in 2019 represents an update of the previous one with the introduction of serum procalcitonin.

The study aims to evaluate the antibiotic prescriptions in patients with community-acquired pneumonia (CAP) before and after both CPs (2015 and 2019).

### Methods

The periods analyzed are seven semesters (one before CP-2015 called PRE period, five post CP-2015 called POST 1–5 and 1 post CP-2019 called POST6).

**Funding:** The author(s) received no specific funding for this work.

**Competing interests:** The authors have declared that no competing interests exist.

**Abbreviations:** CAP, Community-Acquired Pneumonia; CAP-ICD9, Community-Acquired pneumonia identified with ICD9-CM codes; CP, Clinical pathway; DOT, Days of therapy; ICD9-CM, International Classification of Diseases, 9th Revision, Clinical Modification; IQR, Interquartile Range; LOT, Length of therapy; LRTI, Lower Respiratory Tract Infection; PED, Pediatric Emergency Department; SD, Standard Deviation.

The patients have been split into two groups: (i) children admitted to the Pediatric Acute Care Unit (INPATIENTS), and (ii) patients evaluated in the PED and sent back home (OUT-PATIENTS). We have analyzed all descriptive diagnosis of CAP (the assessment of episodes with a descriptive diagnosis were conducted independently by two pediatricians) and CAP with ICD9 classification. All antibiotic prescriptions for pediatric patients with CAP were analyzed.

## Results

A drastic reduction of broad-spectrum antibiotics prescription for inpatients has been noticed; from 100.0% in the PRE-period to 66.7% in POST1, and up to 38.5% in POST6. Simultaneously, an increase in amoxicillin use from 33.3% in the PRE-period to 76.1% in POST1 (p-value 0.078 and 0.018) has been seen.

The outpatients' group's broad-spectrum antibiotics prescriptions decreased from 54.6% PRE to 17.4% in POST6. Both for outpatients and inpatients, there was a decrease of macrolides.

The inpatient group's antibiotic therapy duration decreased from 13.5 days (PRE-period) to 7.0 days in the POST6. Antibiotic therapy duration in the outpatient group decreased from 9.0 days (PRE) to 7.0 days (POST1), maintaining the same value in subsequent periods. Overlapping results were seen in the ICD9 group for both inpatients and outpatients.

## Conclusions

This study shows that CPs are effective tools for an antibiotic stewardship program.

Indeed, broad-spectrum antibiotics usage has dropped and amoxicillin prescriptions have increased after implementing the CAP CP-2015 and the 2019 update.

## Introduction

Lower respiratory tract infection (LRTI) is a leading cause of morbidity and mortality in children and adolescents worldwide [1–3]. Although bacterial etiology occurs in 33–70% of community-acquired pneumonia (CAP) [4,5], too many patients still receive antibiotics for viral pneumonia or broad-spectrum antibiotics for uncomplicated bacterial pneumonia.

The Italian antimicrobial prescription rate is one of the highest in Europe, with direct consequences on antibiotic resistance that has become a serious health threat with high healthcare and social costs [6]. A reduction of antibiotic exposure could have an impact on antibiotic consumption and the development of antibiotic resistance worldwide [7].

Pediatric Emergency Departments (PEDs) have a crucial role in LRTI treatment and can drive antimicrobial prescriptions both for inpatients and outpatients.

In the PED setting, the main challenges are high turnover rates for both patients and practitioners, the need for rapid decision-making, and diagnostic uncertainty in empiric prescription [8,9].

Therefore, antibiotic stewardship programs and clinical pathways (CPs) are winning solutions because they have proven to be effective tools for diagnostic and therapeutic decisions [9–14].

In 2015 we created a CP for CAP intending to decrease overall prescription of antibiotics, especially broad-spectrum.

Many studies have shown that biomarkers could help diagnose bacterial infection and guide the initiation, continuation, and escalation of antibiotics. For this reason, in 2019, we implemented the former CP integrating the serum biomarkers procalcitonin (PCT) and C-reactive protein (CPR) breakpoints for the management of LRTI [15,16].

This study evaluates sustainability over six years of a CP implemented in the PED and the impact of the updated CP on antibiotic treatments and therapy duration.

## Materials and methods

### Study design

The study was set at the PED of the Department for Women and Children's Health at Padua University-Hospital. The hospital provides primary and secondary care to a metropolitan area of 350,000 people (45,000 youngers than 15 years) and tertiary care to a regional and extra-regional population, with approximately 25,000 PED visits per year and an overall hospital admission rate from PED of around 7 out of 100 visits.

Children with moderate-severe CAP are usually admitted to the Pediatric Acute Care Unit (PACU), an acute care unit near the PED, which shares the same medical staff.

This quasi-experimental study assessed variations in antibiotic treatments for CAP over six semesters: one preceding 2015-CP implementation (Pre: 15th October 2014–15th April 2015) while the other six after 2015-CP implementation (Post1: 15th October 2015–15th April 2016, Post2: 15th October 2016–15th April 2017, Post3: 15th October 2017–15th April 2018, Post4: 15th October 2018–15th April 2019, Post5: 15th October 2019–15th December 2019, Post6: 16th December 2019–15th April 2020).

### Intervention

The CP is a one-page decision support algorithm summarizing guidelines and designed to help providers define whether an antibiotic should be prescribed and the optimal agent, and treatment duration.

The CP summarizes international guidelines for the diagnosis and treatment of a specific clinical condition. The first edition was developed in 2015 by the Division of Pediatric Infectious Diseases and the PED of Padua in collaboration with the Division of Pediatric Infectious Diseases of the Children's Hospital of Philadelphia.

A new version of the CP (Figs 1 and 2) was developed and released in December 2019. The main modification was introducing the PCT/CRP guided algorithm designed to guide prescribers about antibiotics treatments in terms of type and therapy duration. PCT was added to the routine blood tests with no need for a new blood sampling and no delay in results turnaround.

CPs were presented to PED and PACU physicians and residents. Yearly training sessions and lectures were organized in order to explain the guidelines and their rationale. CPs pocket cards were delivered to all physicians, and CPs posters were hung on PED walls to make the consultation more immediate.

### Study population and case definition

All consecutive patients aged between 3 months to 14 years evaluated in the PED with an International Classification of Diseases, 9th Revision, Clinical Modification (ICD9-CM: 485, 486) code, or descriptive diagnosis of CAP were included in the study.

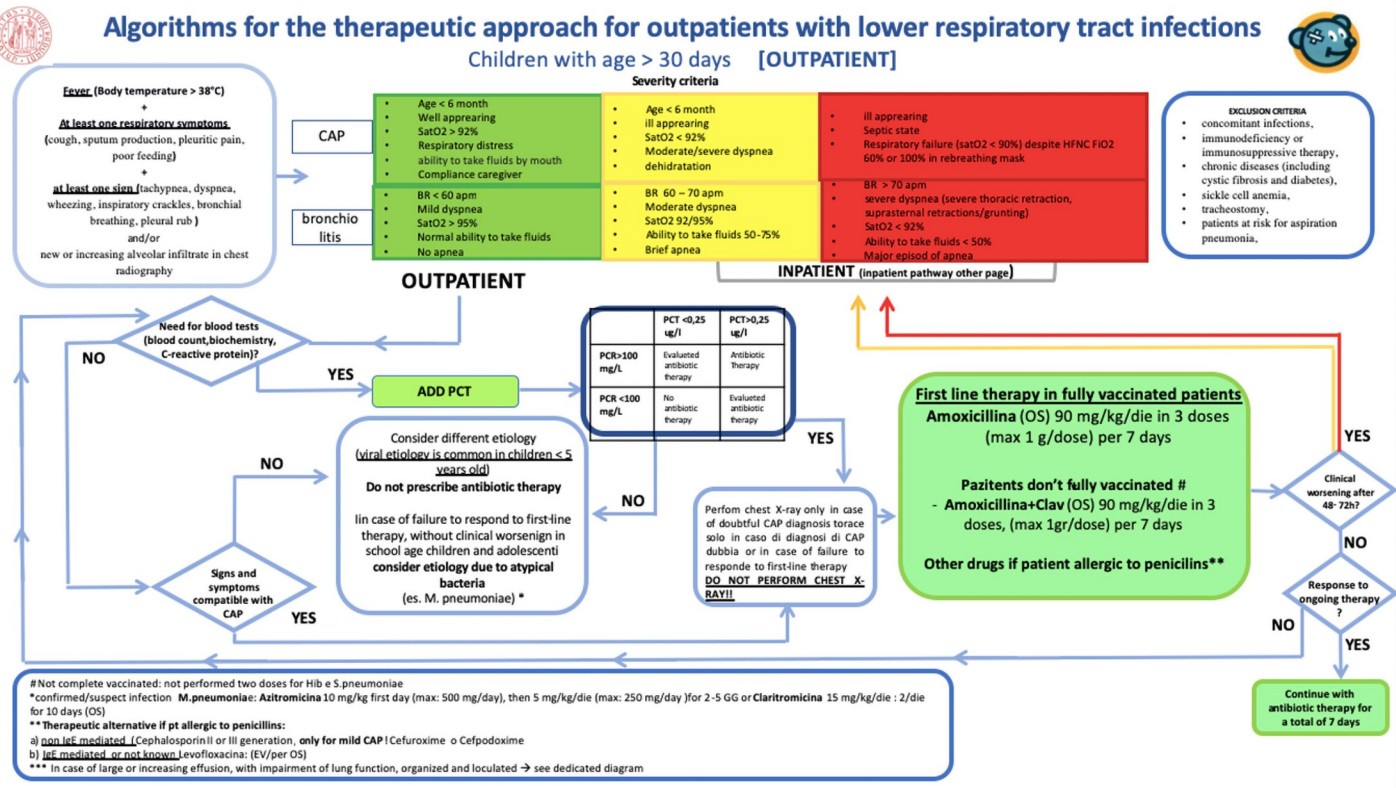

**Fig 1. Clinical pathways algorithms for the therapeutic approach for outpatients with lower respiratory tract infections.** University of Padua, 2019.

Assessments of episodes with a descriptive diagnosis were conducted independently by two pediatricians (SR and FM). Any disagreement was resolved by discussion with a third infectious disease pediatrician (DD).

In general, CAP was defined as the presence of i) fever at the moment of PED visit (core body temperature > 38°C), AND ii) at least one symptom (cough, sputum production, pleuritic pain, poor feeding), AND iii) at least one sign (tachypnea, dyspnea, wheezing, inspiratory crackles, bronchial breathing, pleural rub) for less than 14 days, AND/OR iv) new or increasing alveolar infiltrate in chest radiography [7].

General exclusion criteria were concomitant infections, ongoing antibiotic therapy, immunodeficiency or immunosuppressive therapy, chronic diseases (including cystic fibrosis and diabetes), sickle cell anemia, tracheostomy, patients at risk for aspiration pneumonia, intravenous chemotherapy, wound care, hemodialysis within 30 days of diagnosis, and hospital-acquired pneumonia (pneumonia that occurs 48 hours or more after admission and did not appear to be incubating at the time of admission).

Participating patients were divided into i) outpatients, defined as patients evaluated at the PED and discharged within 6 hours; ii) inpatients, defined as patients admitted to the PACU.

Admissions for CAP occurring in the same patient greater than 30 days apart were analyzed as separate events.

**Data sources and outcomes.**   All clinical, demographic, diagnostic, and prescription data were manually collected from electronic medical records, using a password protected REDCap 10.0.1—© 2020 (Vanderbilt University) data collection form and stored in the secured server at the University of Padua.

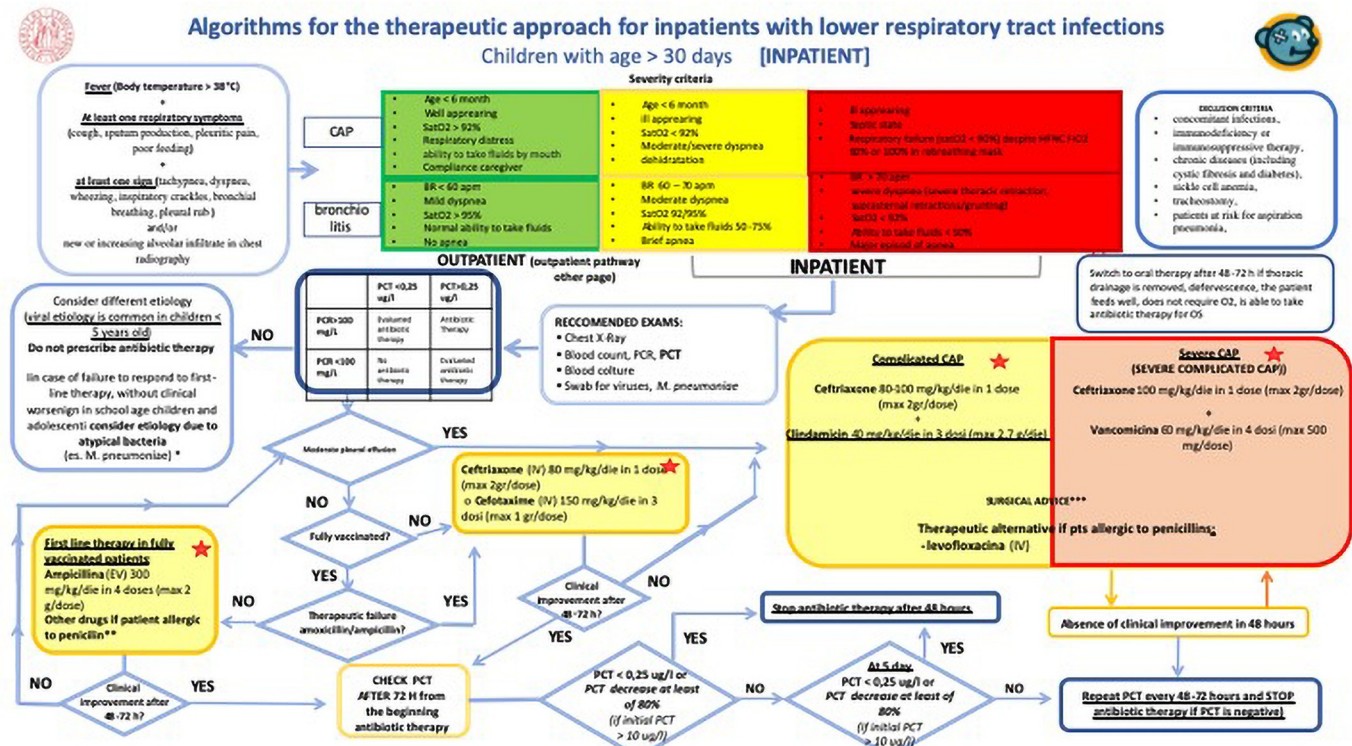

**Fig 2. Clinical pathways algorithms for the therapeutic approach for intpatients with lower respiratory tract infections.** University of Padua, 2019.

We considered treatment based on amoxicillin or ampicillin alone as narrow-spectrum. Broad-spectrum antimicrobials were defined as β-lactams and β-lactamase inhibitor combinations (co- amoxiclav and amoxicillin-sulbactam), second and third-generation cephalosporins, clindamycin, glycopeptides (vancomycin and teicoplanin), fluoroquinolones, and macrolides (azithromycin). Therapeutic regimens, including at least one broad-spectrum prescription, were considered broad-spectrum despite the association with amoxicillin. The CP suggested amoxicillin of 90 mg/kg/day divided every 8 hours as first-line treatment, in line with CAP guidelines [7].

The following aspects of the antibiotic prescriptions for CAP were assessed:

- Broad-spectrum antibiotic treatment (BS-AT) rate;

- Duration of therapy expressed in Days of therapy (DOT), Length of Therapy (LOT) as well as DOT/LOT ratio

- Length of hospital stay (LOS) for inpatients in days.

A single DOT was calculated for each antimicrobial administered to an individual subject within 24 hours, while LOT was the difference from the treatment starting date to the ending date regardless of the different types of antibiotics received. A DOT/LOT ratio >1 meant that combination therapy was prescribed. LOS was calculated by subtracting the date of discharge from the date of admission.

Privacy was guaranteed by assigning to a unique study-specific number each patient and not collecting personally identifying data.

The study was approved by the Institutional Review Board of Padua University-Hospital (3737/AO/16) and complied with international standards including the declaration of Helsinki of 1975 (https://www.wma.net/what-we-do/medical-ethics/declaration-of-helsinki/), revised in 2013 and good clinical practice. Data extraction did not interfere or influence the treatment of patients. Written informed consent was waived by the Institutional Review Board of Padua University-Hospital given the observational nature of the study.

## Data analysis

Results in the different periods were summarized as numbers and percentages (categorical variables) and as median ad interquartile ranges (IQR) (continuous variables). Categorical variables were compared with χ2 or Fisher's 2-tailed exact test in a contingency table and continuous variables with the non-parametric Kruskal-Wallis rank-sum test. Interrupted time series (ITS) analysis supposing an abrupt step change in bi-monthly significative outcomes using a quasi-Poisson regression model was used to determine the intervention's effect.[14] BS-AT and log-transformed total treatments were considered together with a variable representing the frequency in months with which observations were taken and a dummy variable representing CPs implementations. A seasonal adjustment was not necessary since the same calendar months were considered for all the periods to control for effects. Autocorrelation was assessed by examining the plot for residuals and the partial autocorrelation function. Because of the limited sample size in the CAP inpatient group and CAP-ICD9 group, it was not possible to conduct an ITS analysis to determine BS-AT variation. Generalized linear regression models were used to determine variation in therapy duration (LOT and DOT), the length of hospital stay (LOS), and the DOT/LOT ratio in the different periods. Sensitivity analysis was conducted for episodes identified just with ICD9-CM codes (CAP-ICD9). The corresponding relative risk (RR) and 95% confidence interval (95% CI) were calculated for all the models. Data were analyzed using R statistical software (version 3.6.3, Vienna, Austria) for Mac.[17] Figures were created with the packages "ggplot2" [19] and "ggstatsplot" [20]. Statistical significance was set at the 0.05 level, and p values were two-sided.

## Results

In the study period, 1400 CAP episodes were assessed, of which 384 did not meet inclusion criteria. The episodes' inclusion process is described in Fig 3.

## Inpatients

In total, 161 out of 1016 CAP episodes were included in the inpatients' group; the demographic characteristics of children included were similar for sex and age (Table 1).

All CAP episodes except three received antibiotic treatment. There was a significant variation in the BS-AT rate, from 100% in the Pre period to around 60% until the Post5- period. After introducing 2019-CP (Post6), the BS-AT rate decreased to 38.5% (Table 2). The rate of macrolides treatments varied significantly (p-value <0.001) from 66.7% before 2015-CP implementation to 0% after 2019-CP implementation. Cephalosporins use varied as well (from 77.8% Pre to 38.5% Post6), but the difference was not statistically significant (Table 2).

With a variation from 13.5 to 9.0 to 7.0 days, DOT was significantly lower in the Post1, Post3, and Post6 compared to the Pre-period by around 40%. (Table 2 and S2 Table).

LOT was quasi-significantly lower in the Post1 (-34%) and Post6 (-38%) periods with respect to Pre-period. DOT/LOT ratio was from 20 to 30% lower in the Post-periods compared to the Pre. The pre-intervention median LOT was 9.5, while post-intervention was 9 in

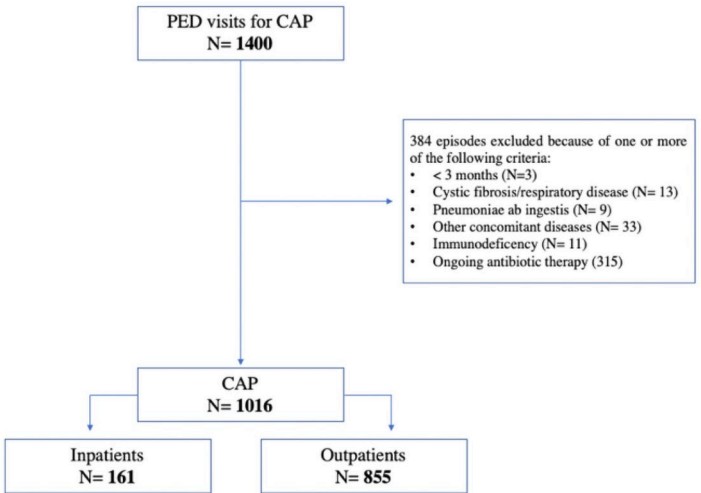

**Fig 3. Flow chart of the episodes inclusion process.**

Post1. From Post2 to Post4, LOT remained constant. In Post6, there was an important LOT decline to 7.0.

LOS was more than 40% lower in the Post-1 and Post3-periods than the Pre-period with a quasi-significant negative difference with the Post6-period (S2 Table).

## Outpatients

Outpatients with CAP included in the study were 855; the demographic characteristics of children included were similar with respect to sex and age (Table 3).

All CAP episodes except 13 received antibiotic treatment. Amoxicillin rate increased from 55.3% to 82.6% after 2019-CP introduction with significant variation between all periods (Table 4).

**Table 1. Demographic characteristics of patients hospitalized for CAP in the different periods.**

| | Pre | Post1 | Post2 | Post3 | Post4 | Post5 | Post6 | p-value |
|---|---|---|---|---|---|---|---|---|
| **N of episodes** | 18 | 21 | 33 | 35 | 30 | 10 | 14 | |
| **Age, months, Median (IQR)** | 34.00 (30.25) | 39.00 (37.00) | 40.00 (69.00) | 27.00 (44.50) | 30.00 (39.00) | 20.50 (14.50) | 23.00 (15.00) | |
| **Age class, 3–35 months, N, (%)** | 10 (55.6%) | 8 (38.1%) | 16 (48.5%) | 20 (57.1%) | 17 (56.7%) | 8 (80.0%) | 11 (78.6%) | |
| **Sex, male, (%)** | 13 (72.2%) | 9 (42.9%) | 21 (63.6%) | 18 (51.4%) | 13 (43.3%) | 8 (80.0%) | 7 (50.0%) | |
| **Body weight, kg, Median (IQR)** | 13.50 (5.05) | 13.00 (7.50) | 14.00 (11.80) | 13.00 (7.75) | 12.50 (5.88) | 12.00 (1.82) | 10.50 (4.62) | |
| **Vaccination, Yes, (%)** | 17 (94.4%) | 20 (95.2%) | 32 (97.0%) | 35 (100.0%) | 30 (100.0%) | 10 (100.0%) | 13 (92.9%) | |
| Not completed the immunization plan | 0 (0.0%) | 0 (0.0%) | 6 (18.8%) | 4 (11.8%) | 1 (3.4%) | 1 (10.0%) | 2 (15.4%) | |
| **C reactive protein exam, Yes, (%)** | / | / | / | / | 29 (96.7%) | 10 (100.0%) | 14 (100.0%) | |
| mg/L, Median (IQR) | / | / | / | / | 47.30 (130.60) | 189.45 (169.32) | 27.10 (19.55) | |
| **Procalcitonin, Yes, (%)** | / | / | / | / | 9 (30.0%) | 5 (50.0%) | 14 (100.0%) | |
| ug/L, Median (IQR) | / | / | / | / | 2.43 (3.63) | 3.75 (4.04) | 0.89 (2.68) | |
| ***M. pneumoniae* positivity** | 3 (16.7%) | 2 (9.5%) | 1 (3.0%) | 2 (5.7%) | 4 (13.3%) | 0 (0.0%) | 0 (0.0%) | |
| **Chest X-ray exams, Yes, (%)** | 17 (94.4%) | 21 (100.0%) | 31 (93.9%) | 32 (91.4%) | 28 (93.3%) | 10 (100.0%) | 14 (100.0%) | |
| **Antibiotic administered** | 18 (100.0%) | 21 (100.0%) | 33 (100.0%) | 34 (97.1%) | 29 (96.7%) | 10 (100.0%) | 13 (92.9%) | |

Only significant p-values are reported.

**Table 2. Antibiotic treatments for patients hospitalized with CAP.**

| | Pre | Post1 | Post2 | Post3 | Post4 | Post5 | Post6 | p-value |
|---|---|---|---|---|---|---|---|---|
| **N of episodes** | 18 | 21 | 33 | 34 | 29 | 10 | 13 | |
| Amoxicillin, N, (%) | 6 (33.3%) | 16 (76.2%) | 14 (42.4%) | 17 (50.0%) | 18 (62.1%) | 7 (70.0%) | 8 (61.5%) | |
| Amikacin, N, (%) | 0 (0.0%) | 0 (0.0%) | 0 (0.0%) | 0 (0.0%) | 1 (3.4%) | 0 (0.0%) | 0 (0.0%) | |
| Co-amoxiclav, N, (%) | 2 (11.1%) | 1 (4.8%) | 7 (21.2%) | 8 (23.5%) | 5 (17.2%) | 0 (0.0%) | 3 (23.1%) | |
| Beta-lactams inhibitors N, (%) | 4 (22.2%) | 1 (4.8%) | 7 (21.2%) | 8 (23.5%) | 5 (17.2%) | 0 (0.0%) | 3 (23.1%) | |
| III-gen Cephalosporins, N, (%) | 14 (77.8%) | 10 (47.6%) | 19 (57.6%) | 15 (44.1%) | 9 (31.0%) | 5 (50.0%) | 5 (38.5%) | |
| Macrolides, N, (%) | 12 (66.7%) | 7 (33.3%) | 5 (15.2%) | 6 (17.6%) | 9 (31.0%) | 1 (10.0%) | 0 (0.0%) | < 0.001 |
| Glycopeptides, N, (%) | 4 (22.2%) | 0 (0.0%) | 2 (6.1%) | 1 (2.9%) | 1 (3.4%) | 0 (0.0%) | 0 (0.0%) | |
| Clindamycin, N, (%) | 0 (0.0%) | 1 (4.8%) | 4 (12.1%) | 5 (14.7%) | 2 (6.9%) | 2 (20.0%) | 1 (7.7%) | |
| **Broad-spectrum treatment, N, (%)** | 18 (100.0%) | 14 (66.7%) | 23 (69.7%) | 20 (58.8%) | 17 (58.6%) | 6 (60.0%) | 5 (38.5%) | 0.005 |
| **DOT, Mean (SD)** | 17.06 (9.64) | 9.62 (5.23) | 13.33 (13.66) | 10.24 (5.35) | 12.21 (10.03) | 9.70 (6.07) | 9.15 (6.20) | |
| **DOT, Median (IQR)** | 13.50 (9.25) | 9.00 (3.00) | 9.00 (3.00) | 9.00 (3.75) | 9.00 (5.00) | 9.00 (5.25) | 7.00 (5.00) | |
| **LOT, Mean (SD)** | 10.78 (4.05) | 8.19 (3.12) | 10.24 (5.99) | 8.76 (3.47) | 9.48 (5.27) | 8.60 (4.20) | 7.77 (3.32) | |
| **LOT, Median (IQR)** | 9.50 (1.75) | 9.00 (3.00) | 9.00 (3.00) | 8.00 (3.00) | 8.00 (4.00) | 9.00 (3.50) | 7.00 (4.00) | |
| **DOT/LOT, Mean (SD)** | 1.53 (0.39) | 1.14 (0.26) | 1.17 (0.34) | 1.17 (0.33) | 1.22 (0.32) | 1.09 (0.18) | 1.12 (0.27) | |
| **DOT/LOT, Median (IQR)** | 1.48 (0.59) | 1.00 (0.29) | 1.00 (0.00) | 1.00 (0.15) | 1.00 (0.50) | 1.00 (0.00) | 1.00 (0.00) | |
| **LOS, Mean (SD)** | 8.06 (7.73) | 4.71 (2.92) | 6.33 (6.50) | 4.79 (3.03) | 5.45 (4.12) | 5.00 (2.26) | 4.92 (3.07) | |
| **LOS, Median (IQR)** | 5.00 (3.75) | 4.00 (2.00) | 4.00 (3.00) | 4.00 (2.75) | 4.00 (3.00) | 4.50 (1.75) | 4.00 (3.00) | |

Only significant p-values are reported.

The 2015-CP implementation was significant (p = 0.031) in decreasing bimonthly BS-AT rates by 47.8% (RR: 0.522 (95% CI: 0.289–0.943)) with stable rates in the following periods (Fig 4 and S1 Table).

The cephalosporins and macrolides treatment rate varied significantly from 13.5% to 3.8% and from 34.8% to 9.1% in the Pre- and Post6-period, respectively (Table 4).

**Table 3. Demographic characteristics of outpatient with CAP in the different periods.**

| | Pre | Post1 | Post2 | Post3 | Post4 | Post5 | Post6 | p-value |
|---|---|---|---|---|---|---|---|---|
| **N of episodes** | 142 | 88 | 142 | 148 | 158 | 43 | 134 | |
| **Age, months, Median (IQR)** | 44.50 (36.75) | 39.50 (25.75) | 38.0 (36.75) | 36.0 (33.0) | 38.50 (31.0) | 48.0 (59.0) | 47.0 (57.75) | 0.001 |
| **Age class, 3–35 months, N, (%)** | 50 (35.2) | 35 (39.8) | 65 (45.8) | 73 (49.3) | 71 (44.9) | 15 (34.9) | 50 (37.3) | |
| **Sex, male, (%)** | 82 (57.7) | 42 (47.7) | 79 (55.6) | 78 (52.7) | 91 (57.6) | 25 (58.1) | 77 (57.5) | |
| **Body weight, kg, Median (IQR)** | 15.75 (8.00) | 14.85 (5.95) | 15.00 (8.00) | 14.00 (6.12) | 14.50 (7.75) | 17.00 (12.05) | 16.00 (12.00) | |
| **Vaccination, Yes, (%)** | 134 (94.4) | 83 (94.3) | 135 (95.1) | 142 (95.9) | 156 (98.7) | 42 (97.7) | 131 (97.8) | |
| Not completed the immunization plan | 6 (4.5) | 5 (6.0) | 2 (1.5) | 9 (6.3) | 10 (6.4) | 3 (7.1) | 5 (3.8) | |
| **C reactive protein exam, Yes, (%)** | 24 (16.9) | 8 (9.1) | 11 (7.7) | 19 (12.8) | 26 (16.5) | 9 (20.9) | 17 (12.7) | |
| mg/L, Median (IQR) | 20.50 (48.58) | 17.00 (17.60) | 39.55 (26.10) | 53.00 (53.35) | 52.60 (86.95) | 38.00 (99.10) | 21.50 (94.40) | |
| **Procalcitonin, Yes, (%)** | / | / | / | / | 4 (2.5) | 1 (2.3) | 8 (6.0) | |
| ug/L, Median (IQR) | / | / | / | / | 8.49 (11.52) | 1.09 (0.00) | 0.12 (0.57) | |
| ***M. pneumoniae* positivity** | 3 (2.1) | 0 (0.0) | 1 (0.7) | 0 (0.0) | 0 (0.0) | 0 (0.0) | 0 (0.0) | |
| **Chest X-ray exams, Yes, (%)** | 67 (47.2) | 34 (38.6) | 59 (41.5) | 45 (30.4) | 64 (40.5) | 23 (53.5) | 47 (35.1) | <0.001 |
| **Antibiotic administered** | 141 (99.3) | 88 (100.0) | 142 (100.0) | 145 (98.0) | 153 (96.8) | 41 (95.3) | 132 (98.5) | |

Only significant p values are reported.

**Table 4. Antibiotic treatments for outpatients with CAP.**

|  | Pre | Post1 | Post2 | Post3 | Post4 | Post5 | Post6 | p-value |
|---|---|---|---|---|---|---|---|---|
| **N of episodes** | 141 | 88 | 142 | 145 | 153 | 41 | 132 |  |
| Amoxicillin, N, (%) | 78 (55.3) | 64 (72.7) | 113 (79.6) | 124 (85.5) | 120 (78.4) | 33 (80.5) | 109 (82.6) | < 0.001 |
| Amikacin, N, (%) | 0 (0.0) | 0 (0.0) | 0 (0.0) | 0 (0.0) | 2 (1.3) | 0 (0.0) | 0 (0.0) |  |
| Co-amoxiclav, N, (%) | 23 (16.3) | 10 (11.4) | 13 (9.2) | 16 (11.0) | 11 (7.2) | 4 (9.8) | 7 (5.3) |  |
| Beta-lactams inhibitors, N, (%) | 23 (16.3) | 10 (11.4) | 13 (9.2) | 16 (11.0) | 11 (7.2) | 4 (9.8) | 7 (5.3) |  |
| III gen Cephalosporins, N, (%) | 19 (13.5) | 7 (8.0) | 8 (5.6) | 6 (4.1) | 6 (3.9) | 0 (0.0) | 5 (3.8) | 0.003 |
| Macrolides, N, (%) | 49 (34.8) | 12 (13.6) | 10 (7.0) | 4 (2.8) | 20 (13.1) | 6 (14.6) | 12 (9.1) | < 0.001 |
| Glycopeptides, N, (%) | 0 (0.0) | 0 (0.0) | 0 (0.0) | 0 (0.0) | 0 (0.0) | 0 (0.0) | 0 (0.0) |  |
| Clindamycin, N, (%) | 0 (0.0) | 0 (0.0) | 0 (0.0) | 0 (0.0) | 0 (0.0) | 0 (0.0) | 0 (0.0) |  |
| **Broad-spectrum treatment, N, (%)** | 77 (54.6) | 26 (29.5) | 31 (21.8) | 26 (17.9) | 35 (22.9) | 10 (24.4) | 23 (17.4) | <0.001 |
| **DOT, Mean (SD)** | 9.26 (4.00) | 7.14 (1.91) | 6.61 (1.34) | 6.94 (1.17) | 6.92 (1.84) | 7.34 (2.36) | 6.72 (1.51) |  |
| **DOT, Median (IQR)** | 9.00 (3.00) | 7.00 (2.00) | 6.00 (1.00) | 7.00 (1.00) | 7.00 (1.00) | 7.00 (0.00) | 7.00 (1.00) |  |
| **LOT, Mean (SD)** | 8.21 (2.28) | 6.94 (1.38) | 6.58 (1.31) | 6.87 (0.99) | 6.76 (1.49) | 7.02 (1.54) | 6.70 (1.51) |  |
| **LOT, Median (IQR)** | 9.00 (2.00) | 7.00 (1.25) | 6.00 (1.00) | 7.00 (1.00) | 7.00 (1.00) | 7.00 (0.00) | 7.00 (1.00) |  |
| **DOT/LOT, Mean (SD)** | 1.12 (0.29) | 1.02 (0.13) | 1.00 (0.06) | 1.01 (0.07) | 1.02 (0.11) | 1.04 (0.18) | 1.00 (0.03) |  |
| **DOT/LOT, Median (IQR)** | 1.0 (0.0) | 1.0 (0.0) | 1.0 (0.0) | 1.0 (0.0) | 1.0 (0.0) | 1.0 (0.0) | 1.0 (0.0) |  |

Only significant p-values are reported.

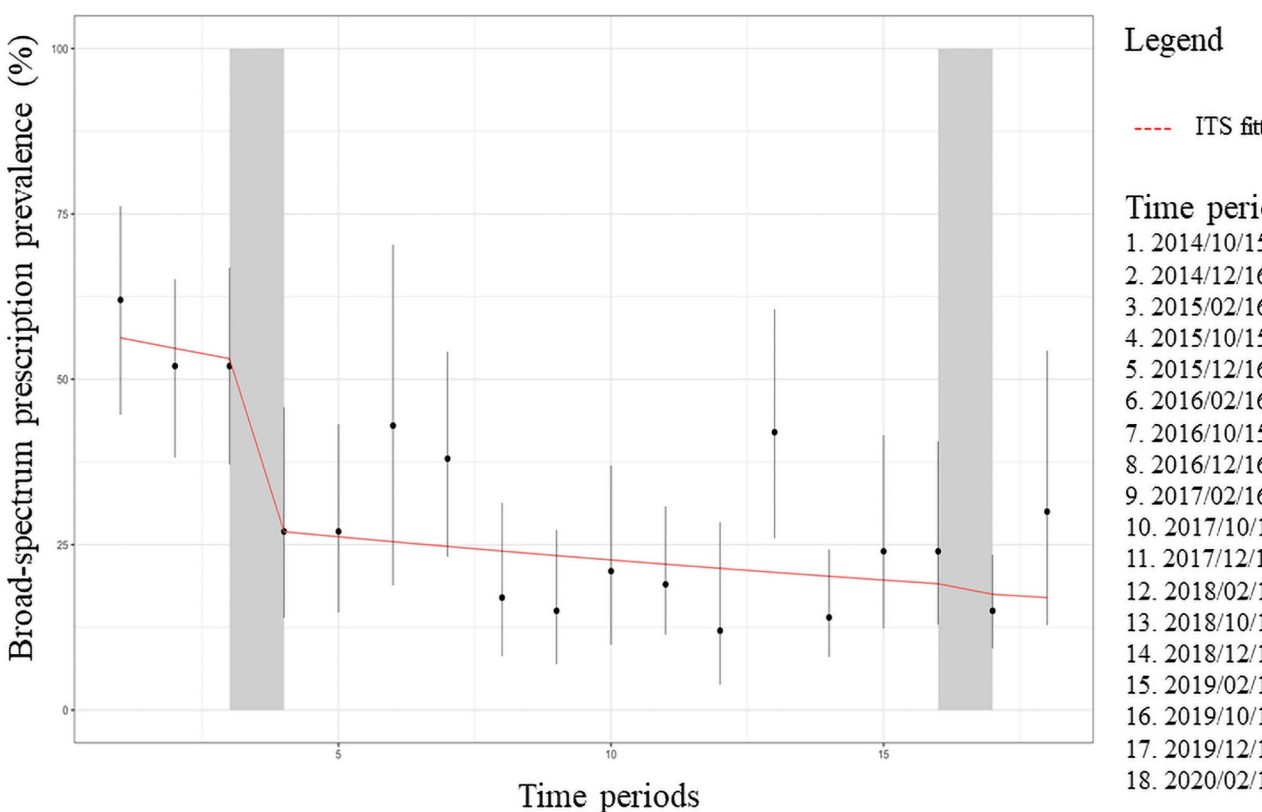

**Fig 4. Interrupted time series of bimonthly broad-spectrum antibiotics prescriptions for outpatients (dots) expressed as percentages with 95% confidence intervals (bars) for community-acquired pneumonia.** The red line represents the broad-spectrum prescriptions trend for inpatients.

DOT and LOT had a similar pattern decreasing from 9.0 days to 7.0 days, and values were about 20% lower in the different post periods compared to the Pre-period (Table 4 and S1 Table). DOT/LOT ratios were significantly lower in all periods post 2015-CP implementation than the pre-implementation ratio except for Post1 and Post5 periods (S1 Table).

**CAP-ICD9.** Secondary analysis on ICD9-CM confirmed CAP (n = 728) confirmed presented results and are included in the **Supporting Information** S1 File. **CAP-ICD9 analysis.**

## Discussion

CPs proved to be a feasible strategy in reducing the use of antibiotics for LRTI in the PED. LRTIs, such as pneumonia, are among the main challenges in children's primary care because of the frequency and the unclear etiology, which are also the causes of excessive prescription of antibiotics.

According to the guidelines [7], the first-line therapy for CAP should be based on amoxicillin alone (90 mg/kg/die for seven days) because of a possible increase in Penicillin Binding Protein mutation in *Streptococcus pneumoniae* causing drug resistance, leaving BS-ATs, such as co-amoxiclav and III generation cephalosporins, the first choice just in non-fully immunized children [17]. In our CP, the use of co-amoxiclav is recommended in not fully immunized patients (to cover H. Influenzae). While in the case of amoxicillin treatment failure, III generation cephalosporins are the drug of choice.

In outpatients, the BS-ATs rate declined just after the 2015-CP implementation with stable rates in the following semesters until the 2019-CP update. The reduced sample size may cause the fact that the decline was not significant in the inpatient cohort since a higher sample size is needed to see a smaller variation in the outcome of interest.

The low usage of amoxicillin before the CP-2015 publication could reflect prescribers' belief that, when there is a severe infection, BS-ATs are more suitable, especially if no follow-up visit is planned. However, this might not be valid in the Italian Healthcare System, where each child is assigned to a primary care pediatrician, which is the primary referral for any health-related matters, thus guaranteeing continuity in the patient's care. Nonetheless, our results align with the literature findings in a similar setting, with a high BS-ATs rate prescribed for LRTI [18,19]. Other key drivers for BS-ATs prescriptions might be represented by the young age and by the lack of pneumococcal vaccination and non-normal laboratory findings [20].

The changes from combination therapy to monotherapy led to a significant reduction in DOT and LOT, even if the inpatients with CAP did not last in the latter. This data could suggest that pediatricians are more prone to change their attitude about the antibiotic choice rather than the therapy duration. Moreover, we observed no variation in *Mycoplasma pneumoniae* rates in different periods, leaving scarce justification for macrolides use.

Interestingly, in the periods immediately before and after 2019-CP, inpatient LOT was quasi-significantly lower than before 2015-CPs implementation value. It may be asserted that when a new policy is implemented, prescribers tend to check the guidelines more often, thus leaving fewer opportunities for mistakes [21].

Reduced LOS in the first and third semesters after 2015-CP could be explained by the change in rate in other factors that led to hospitalization in children, such as dehydration. If more children needed intravenous hydration in a specific period, the LOS would have increased.

This study is an update of preliminary findings published in 2018 by Donà et al. [16], which allowed improving CPs, including serum biomarkers monitoring, to reduce therapy duration. BS-AT reduction was achieved after the first CP implementation, but different studies used the PCT benchmark to reduce antibiotic prescriptions in the literature. For example, in a

prospective cluster randomized controlled trial (RCT) in 243 adult patients with LRTI, PCT-guided treatment algorithm reduced antibiotic exposure by 51% (adjusted relative risk 0.49 (95% CI 0.44–0.55; p < 0.0001)) compared with the standard of care group [22].

Our findings of reduced antibiotic exposure are in line with the literature in both adult and pediatric studies. Schuetz and colleagues reported that PCT guidance was associated with a 2.4-day reduction in antibiotic exposure and a lower risk of antibiotic-related side effects [23]. Similar results were confirmed for pediatric patients in an RCT conducted by Esposito et al.; on hospitalized patients with CAP, the use of an algorithm with a PCT benchmark of 0.25 ng/ml was significant in reducing both antibiotic-related adverse events both antibiotic therapy duration [15].

Another RCT conducted by Baer et al. [16] involving 337 children and adolescents with LRTI presenting to the PED found a reduced antibiotic therapy duration in children treated according to PCT-guided algorithm (−1.8 days for all LRTI and −3.4 days for pneumonia). The fact that higher cut-offs based on adult studies were used limits the comparison with our findings.

Even if costs were not assessed, Mewes et al. [24] found that total cost was reduced by 17.7% in hospitalized patients with LRTI treated according to a PCT algorithm versus standard of care. Indeed, even if PCT and CRP serum measuring impact on overall costs (14.40 EUR for each PCT test and 4.20 EUR for each CRP test), especially for outpatients (considering that a standard visit costs 25.00 EUR), the overall reduction in days of hospitalization should be considered as well.

The economic impact was not the study's goal, but it is the reason for another study currently underway. Turnaround time did not affect prescribing behavior because it is the same time that routine blood tests take.

In line with the literature [25], our study confirmed that a combination of biomarkers is valid to support therapeutic decisions for pediatric LRTI.

This study has points of strength but also some limitations. This intervention was developed by a multidisciplinary team to guarantee a high level of quality and coordination, thanks to the cooperation of the Infectious Diseases and PED teams. Moreover, pathways were easily accessible to prescribers since laminated pocket cards were delivered and included in the hospital policy. As previously noted in other studies, [9,26] he latter played a role in prescribers' compliance with CP. Medical records were reviewed by expert pediatricians to confirm the diagnosis of pneumonia for all children included in the study, and secondary analysis on ICD9-CM confirmed CAP was performed as well.

Limitations include the monocentric quasi-experimental design of the study, even if it is assumed to be the best design when randomization is not possible (such in our care), and the small sample size in some periods that reduced statistical strength. In our favor, we used multiple time points after the intervention to control for underlying trends. The assumption of at least three data points in the time series both before and after the intervention was not met when assessing 2019-CP; hence just the 2015-CP impact was assessed with the ITS. For the same reason, it was not possible to evaluate either 2015-CP either 2019-CP for outpatient with CAP-ICD9, while for inpatient, the reduced sample size in time-series datapoint limited the use of this technique [27,28].

Second, patient follow-up was beyond the aim of the study and was, therefore, not performed. Even if our study was not assessing patients' outcomes prospectively (e.g.. through a phone call), the absence of a second PED admission might suggest no difference nor in treatment failure nor adverse event rate in all periods. Third, the sample size was studied to assess the reduction of BS-AT based on previous PED access. COVID-19 pandemic was reduced by almost 70% PED visit rate during the lock-down period, but still positive results after policy

implementation were observed. Fourth, being a single-center study, our results may not be generalizable to other institutions because factors such as local leadership and culture play an essential role in policy success [28]. Indeed, at Padua University Hospital is in place a stewardship program targeting different conditions. Thus prescribers might be more prone to accept a policy targeting antibiotics treatments [29,30]. Nonetheless, the guidelines, once adapted, and the methodology could be replicated in other centers, possibly leading to similar results.

## Conclusions

CP has shown to be an adequate tool in improving prescribing appropriateness during these six years, but continuous efforts in developing antimicrobial stewardship programs are needed. Other studies should be performed to confirm the crucial role of biomarkers in assessing children with LRTI.

## Supporting information

**S1 Table. Interrupted time series model parameters for outpatients and inpatients.**
(DOCX)

**S2 Table. Relative risk with 95% confidence intervals and p-value of univariate regression models for DOT, LOT, DOT/LOT ratios and LOS.** The reference for all the models is the Pre period. Significant P values are in bold. Quasi significant p values are in italic.
(DOCX)

**S1 File. CAP-ICD9 analysis.**
(DOCX)

## Author Contributions

**Conceptualization:** Sara Rossin, Elisa Barbieri, Carlo Giaquinto, Liviana Da Dalt, Daniele Doná.

**Data curation:** Sara Rossin, Elisa Barbieri, Francesco Martinolli.

**Formal analysis:** Sara Rossin, Elisa Barbieri, Anna Cantarutti, Francesco Martinolli.

**Funding acquisition:** Sara Rossin, Elisa Barbieri.

**Investigation:** Sara Rossin, Elisa Barbieri, Daniele Doná.

**Methodology:** Sara Rossin, Elisa Barbieri, Daniele Doná.

**Project administration:** Sara Rossin, Elisa Barbieri, Daniele Doná.

**Resources:** Sara Rossin, Elisa Barbieri, Daniele Doná.

**Software:** Sara Rossin, Elisa Barbieri, Daniele Doná.

**Supervision:** Sara Rossin, Elisa Barbieri, Carlo Giaquinto, Liviana Da Dalt, Daniele Doná.

**Validation:** Sara Rossin, Daniele Doná.

**Visualization:** Sara Rossin, Daniele Doná.

**Writing – original draft:** Sara Rossin, Elisa Barbieri.

**Writing – review & editing:** Sara Rossin, Elisa Barbieri.

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
