## [Decision Letter · Decision Letter 0]

19 Mar 2021

PONE-D-21-03385

Multistep antimicrobial stewardship intervention on antibiotic prescriptions and treatment duration in children with pneumonia

PLOS ONE

Dear Dr. Rossin,

Thank you for submitting your manuscript to PLOS ONE. After careful consideration, we feel that it has merit but does not fully meet PLOS ONE’s publication criteria as it currently stands. Therefore, we invite you to submit a revised version of the manuscript that addresses the points raised during the review process.

Please revise considering reviewer comments point by point. 

We look forward to receiving your revised manuscript.

Kind regards,

Iddya Karunasagar

Academic Editor

PLOS ONE

Journal Requirements:

3. In your ethics statement in the manuscript and in the online submission form, please ensure that you have discussed whether all data/samples were fully anonymized before you accessed them and/or whether the IRB or ethics committee waived the requirement for informed consent.

If patients provided informed written consent to have data/samples from their medical records used in research, please include this information.

4. In the ethics statement in the manuscript and in the online submission form, please provide additional information about the patient records/samples used in your retrospective study, including:

a) the date range (month and year) during which patients' medical records/samples were accessed;

b) the date range (month and year) during which patients whose medical records/samples were selected for this study sought treatment.

Additional Editor Comments:

Reviewers have suggested a number of minor improvements in the manuscript. Please revise addressing these comments.

Reviewers' comments:

Reviewer's Responses to Questions

**Comments to the Author**

1. Is the manuscript technically sound, and do the data support the conclusions?

Reviewer #1: Yes

Reviewer #2: Yes

2. Has the statistical analysis been performed appropriately and rigorously? 

Reviewer #1: Yes

Reviewer #2: Yes

3. Have the authors made all data underlying the findings in their manuscript fully available?

Reviewer #1: Yes

Reviewer #2: Yes

4. Is the manuscript presented in an intelligible fashion and written in standard English?

Reviewer #1: Yes

Reviewer #2: Yes

5. Review Comments to the Author

Reviewer #1: The manuscript titled " Multi step antimicrobial stewardship intervention on antibiotic prescriptions and treatment duration in children with pneumonia

The paper does look well into the problem and is well written. The following observations are in order that may require addressal

1. Materials and Methods ( ( Study population and Case definition) :

There is no reference for the criteria that define CAP .

2. Data sources and outcomes : The line " The CP suggestedamoxicillin of ..... guidelines . There is no reference for standard guidelines used

3. Table 1: Percentages have been calculated for denominator s less than 20 ( ideally 30) and this may lead to spuriously higher percentages

4. Table - 2 : There are two categories mentioned against Beta lactams inhibitors , One is Co-amoxiclav . The second category of Beta lactams inhibitors does not state which inhibitors were used . This is not explained in the text as well and the same holds good for 4

5. If we compare the Table -1 and 3 , there is a category that alludes to those children who have not completed the immunisation plan. However the discussion section does not talk about the impact that this may have had on the development of CAP or the requirement of hospitalisation and the final outcome ( the LOT, LOS AND DOT ) in patients who either were sent home from the PED or were admitted to the hospital for CAP. . This needs some discussion

6. Table 2 & 4 The authors have not specified the generation of cephalosporins used for treating children with CAP in both the categories. This is important as it does have an impact on the management of CAP Vs HAP

7. Discussion : The second paragraph describes the mechanism of PRP as the reason for first line therapy empiric , and calls for Co- amoxiclav as the second choice. A S. pneumomiae isolate that develops penicillin resistance may not be well managed even with Amoxycillin clavulanic acid.

Reviewer #2: Dear Authors

The paper gives a clear picture of the impact of clinical pathway and AMS on treatment duration on a paediatric hospital. It is a good study that has been well described by the authors.

I request the authors to explain the following

1. What were the cost implications of the use of procalcitonin in your set up? Especially in out-patient settings. What was the turnaround time for results in the out patient settings that influenced the decision of antimicrobial prescribing behavior?

2. In the methodology, each phase of the study was 6 months (October to April). But in 2019, the phase was split due to the new intervention being introduced. However, if we look at the out patient numbers in each year, there was a dip in 2015-2016 (88 vs an average of 148 patients) and during the time period of October 2019 to April 2020 ( though split as 2 phases), the total number that visited the OPD was 177( 43+134). Could the authors explain the increase or the decrease during these years? What was the incidence of Influenza those years?

3. What were the beta lactam, beta lactamase combinations or macrolides or glycopeptides or fluoroquinolones used?

4. The authors explain that the increase in the use broad spectrum antibiotics in the pre-2nd intervention stage is due to a reduced sample size. Could that also be the reason for the days of therapy and the length of therapy being increased in this period.

6. PLOS authors have the option to publish the peer review history of their article (what does this mean?). If published, this will include your full peer review and any attached files.

Reviewer #1: No

Reviewer #2: No

---

## [Author Response · Author response to Decision Letter 0]

16 Jun 2021

Dear Editors,

Dear Reviewers,

We would like to thank you for the helpful comments and suggestions. We have included a translation of the CP since we believe some comments arose because CP figures were in Italian. We are resubmitting our manuscript after addressing point-by-point all the comments.

Sincerely,

Sara Rossin, on behalf of all the Authors

Reviewer 1

The manuscript titled " Multi step antimicrobial stewardship intervention on antibiotic prescriptions and treatment duration in children with pneumonia. The paper does look well into the problem and is well written. The following observations are in order that may require addressal. 

1. Materials and Methods ( Study population and Case definition) :

There is no reference for the criteria that define CAP.

We thank the reviewer for this comment. We have added the following reference used to define CAP criteria in the appropriate section.

Bradley JS, Byington CL, Shah SS, Alverson B, Carter ER, Harrison C, et al. Pediatric Infectious Diseases Society and the Infectious Diseases Society of America: The Management of Community- Acquired Pneumonia in Infants and Children Older Than 3 Months of Age: Clinical Practice Guidelines by the Pediatric Infectious Diseases Society of America. Clin Infect Dis 2011; 53(7): e25–76 https://doi. org/10.1093/cid/cir531 PMID: 21880587 

2. Data sources and outcomes : The line " The CP suggested amoxicillin of ..... guidelines . There is no reference for standard guidelines used. 

We thank the reviewer for this comment. We have added the following reference used to define the treatment for CAP in the CP. The 2019-CP the referred guidelines were the same as the 2015-CP since no update on the antibiotic treatment was published.

Bradley JS, Byington CL, Shah SS, Alverson B, Carter ER, Harrison C, et al. Pediatric Infectious Diseases Society and the Infectious Diseases Society of America: The Management of Community- Acquired Pneumonia in Infants and Children Older Than 3 Months of Age: Clinical Practice Guidelines by the Pediatric Infectious Diseases Society of America. Clin Infect Dis 2011; 53(7): e25–76 https://doi. org/10.1093/cid/cir531 PMID: 21880587 

3. Table 1: Percentages have been calculated for denominators less than 20 ( ideally 30) and this may lead to spuriously higher percentages

Thank you for this comment, we agree with the reviewer. For this reason, we reported in the first line of table 1 the numbers of patients used as a denominator for each period. The limited number of patients in some periods was reported as a limitation in the discussion section. 

4. Table - 2 : There are two categories mentioned against Beta lactams inhibitors , One is Co-amoxiclav . The second category of Beta lactams inhibitors does not state which inhibitors were used . This is not explained in the text as well and the same holds good for 4. 

We thank the reviewer for the comment. We have specified in Table 2, Table 4, and the manuscript the different molecules for each antibiotic category.

5. If we compare Table -1 and 3 , there is a category that alludes to those children who have not completed the immunisation plan. However the discussion section does not talk about the impact that this may have had on the development of CAP or the requirement of hospitalisation and the final outcome ( the LOT, LOS AND DOT ) in patients who either were sent home from the PED or were admitted to the hospital for CAP. This needs some discussion. 

We thank the reviewer for the suggestion. Children not vaccinated received broad-spectrum antibiotics in line with our CPs and the guidelines reported in the references. The population of children that were not fully vaccinated did not differ in the various periods. Moreover, the decision algorithm for hospitalization is not based on the immunization status but age (> or < 6 months) and the signs and symptoms. We think it could not be helpful to compare outcomes in outpatients with patients hospitalized for pneumonia since the CP is different. 

6. Table 2 & 4 The authors have not specified the generation of cephalosporins used for treating children with CAP in both the categories. This is important as it does have an impact on the management of CAP Vs HA. 

We thank the reviewer for the comment. We acknowledge that different types of antibiotics are used in the management of HAP and CAP. 

The CP has been explicitly designed for community-acquired LRTI. Hence we did not include HAP episodes in the study.

Furthermore, we have now specified in the text the class and the type of cephalosporines suggested by the CP and prescribed (III generation).

7. Discussion : The second paragraph describes the mechanism of PRP as the reason for first line therapy empiric , and calls for Co- amoxiclav as the second choice. A S. pneumomiae isolate that develops penicillin resistance may not be well managed even with Amoxycillin clavulanic acid.

Thank you for this comment. In our Region, the rate of penicillin-resistant S . pneumoniae isolates in children is low compared to other Regions in Italy or other European countries. We agree with the reviewer that a penicillin-resistant S. pneumoniae is not well managed with co-amoxiclav, and we specified it in the discussion section. In our CP, the use of co-amoxiclav is recommended in not fully immunized patients (to cover H. Influenzae). While in the case of penicillin/amoxicillin treatment failure, ceftriaxone is the drug of choice. 

Reviewer #2: Dear Authors

The paper gives a clear picture of the impact of clinical pathway and AMS on treatment duration on a paediatric hospital. It is a good study that has been well described by the authors.

I request the authors to explain the following

1. What were the cost implications of the use of procalcitonin in your set up? Especially in out-patient settings. What was the turnaround time for results in the outpatient settings that influenced the decision of antimicrobial prescribing behavior?

We thank the reviewer for the comment. The economic impact was not the study's goal, but it is the aim of another study currently underway. The turnaround time of PCT exams is the same as other tests usually performed in the ER. For this reason, this test could change prescribing behavior of physicians helping them distinguish possible bacterial infections in the same amount of time. 

2. In the methodology, each phase of the study was 6 months (October to April). But in 2019, the phase was split due to the new intervention being introduced. However, if we look at the out patient numbers in each year, there was a dip in 2015-2016 (88 vs an average of 148 patients) and during the time period of October 2019 to April 2020 ( though split as 2 phases), the total number that visited the OPD was 177( 43+134). Could the authors explain the increase or the decrease during these years? What was the incidence of Influenza those years?

Thank you for the comment. During the study period, the proportion of LRTI pediatric emergency department visits on overall PED visits was constant, with a peak between December and January of each study period (see Figure 1 below). PED Visits related to flu were higher between 2015-2020 compared to 2014-15, while monthly total PED visits were stable across all study periods. (see Figure 2 below). Please note that the data presented in the Figures below represent overall PED visits, and no exclusion was performed based on our study exclusion criteria. A possible explanation for the different sizes in outpatient numbers in 2015-2016 might be linked to the differences in eligibility criteria for CP use. In particular, the CP does not apply to children visited in the PED already taking an antibiotic treatment prescribed previously by their primary care pediatrician. We would like to highlight that all episodes with a descriptive diagnosis were independently assessed by two pediatricians (SR and FM). Any disagreement was resolved by discussion with a third infectious disease pediatrician (DD) as stated in the Methods part – Study population and case definition.

---

## [Decision Letter · Decision Letter 1]

16 Sep 2021

Multistep antimicrobial stewardship intervention on antibiotic prescriptions and treatment duration in children with pneumonia

PONE-D-21-03385R1

Dear Dr. Rossin,

We’re pleased to inform you that your manuscript has been judged scientifically suitable for publication and will be formally accepted for publication once it meets all outstanding technical requirements.

Kind regards,

Iddya Karunasagar

Academic Editor

PLOS ONE

Additional Editor Comments (optional):

All comments have been addressed.

Reviewers' comments:

Reviewer's Responses to Questions

**Comments to the Author**

1. If the authors have adequately addressed your comments raised in a previous round of review and you feel that this manuscript is now acceptable for publication, you may indicate that here to bypass the “Comments to the Author” section, enter your conflict of interest statement in the “Confidential to Editor” section, and submit your "Accept" recommendation.

Reviewer #2: (No Response)

2. Is the manuscript technically sound, and do the data support the conclusions?

Reviewer #2: Yes

3. Has the statistical analysis been performed appropriately and rigorously? 

Reviewer #2: Yes

4. Have the authors made all data underlying the findings in their manuscript fully available?

Reviewer #2: Yes

5. Is the manuscript presented in an intelligible fashion and written in standard English?

Reviewer #2: Yes

6. Review Comments to the Author

Reviewer #2: Given the aims and objectives of the study the authors have sufficiently addressed all queries. The manuscript may be accepted

7. PLOS authors have the option to publish the peer review history of their article (what does this mean?). If published, this will include your full peer review and any attached files.

Reviewer #2: **Yes: **Anusha Rohit

---

## [Editor Report · Acceptance letter]

24 Sep 2021

PONE-D-21-03385R1 

Multistep antimicrobial stewardship intervention on antibiotic prescriptions and treatment duration in children with pneumonia 

Dear Dr. Rossin:

I'm pleased to inform you that your manuscript has been deemed suitable for publication in PLOS ONE. Congratulations! Your manuscript is now with our production department. 

Kind regards, 

on behalf of

Dr. Iddya Karunasagar 

Academic Editor

PLOS ONE